# Dimensionless Pressure Response Analysis for Water Supply Pipeline Systems with or without Pumping Station

Sanghyun Kim

Department of Environmental Engineering, Pusan National University, Busan 46241, Republic of Korea;
kimsangh@pusan.ac.kr; Tel.: +82-51-510-2479

**Abstract:** Dimensionless governing equations of unsteady flow and solutions for the pipeline systems having a surge tank were developed. Frequency domain pressure response expressions for two widely used water supply systems were analyzed in dimensionless frequency and time domains. One is the simple reservoir pipeline surge tank valve system and the other is the pipeline system with pump and check valve protected by surge tank. Two different dimensionless expressions for the surge tank were developed and the performance of two expressions was compared. The frequency response pattern of impedance at the downstream valve indicated that the system resonance was determined by the lengths of the main pipeline and the connector and the locations of the surge tank and check valve for the protection of the pumping station. The difference between the simple pipeline system and that of a pump with a check valve was expressed in terms of the phase difference in frequency response distribution. The integrated pressure response for the protected pipeline section was evaluated considering the impact of surge protection in the frequency domain. A better understanding of system behavior can be obtained in terms of specific component isolation both in the frequency and time domain pressure responses. The driven responses in the frequency and time domain can be an important basis for optimum design and operation conditions of water supply systems in dimensionless space.

**Keywords:** water hammer; pipelines with surge tank; dimensionless responses; pump with check valve

## 1. Introduction

Water hammers in pipeline systems had been an important engineering problem due to burst, leakage, and water quality issues. Transient events can be generated from sudden valve maneuvers, and the abrupt action of pump and check valve reactions in pipeline systems. Pressure waves can originate from abrupt changes in the flow velocity, which introduce either overpressure or low pressure along the pipeline. Although high pressure can burst a weakened section of the pipeline, low pressure can generate column separation and cavitation, thus severely damaging the pipeline system. To protect pipeline structures from hydraulic transient events, surge protection devices such as surge tanks and air chambers have been widely used in front of control valves and pump stations. Therefore, the evaluation of water hammer events is a critical requirement for the proper design of pipeline surge arrest devices such as the surge tank and air chamber. The impact of surge pressure depends on various factors such as pipeline dimensions, layout, transient introducing conditions, and various hydraulic structures (e.g., valves and pumps). Substantial works have been explored to address many factors to consider each particular system feature.

To analyze surge events in pipeline systems, the characteristic method (MOC) has been [1–4]. Additionally, the size and location of surge protection devices have been determined to relax abrupt pressure variations, while considering the cost of surge protection [5–10]. Many studies showed that the control of water hammers or the design of surge tanks in pipeline systems can be performed based on an MOC basis [11,12].

Pressure wave generation and its propagation along the pipeline system can introduce pressure oscillations, which can be expressed by the surface water variation of the surge tank [13]. The length of the main pipeline and the location, cross-sectional area, and connector length of the surge tank in the pipeline system are important variables for determining the resonance characteristics of the pressure response. The application of the impedance method demonstrates the potential of the frequency domain approach in the context of resonance characterization for the design of hydraulic structures via transient analysis as well as its complementary time domain response [14]. Assuming that the layouts of pipeline systems with a surge tank are simple and similar to each other (e.g., the surge tank valve of a reservoir pipeline), the pressure wave propagation pattern of the pipeline system can be generalized via the dimensionless development of governing equations and its optimum solution in the time domain response [15]. Actually, the dimensionless approach can substantially generalize the characteristics of the pressure response pattern if the layout of the pipeline system is identical. In other words, if the general response pattern in dimensionless space is determined, then simple conversion into real dimension provides numerous representations of analysis as long as the structure of the pipeline system is identical.

Time domain responses of dimensionless solutions provided a general guideline for the design of surge tanks. However, a dimensionless analysis based on the pressure response must be performed in the frequency domain to configure the interaction of each pipeline segment and component, which allows one to comprehensively understand the system behavior and to conduct better management for various transient scenarios. Furthermore, a more general layout of pipeline systems, such as a pipeline system with pumping stations and check valves, must be developed and compared between systems with/without a specific component to provide a holistic evaluation of the pressure response in regard to impact identification for specific events. Actually, computational fluid dynamic analyses have been performed to improve the design of centrifugal pumps using multiple objective optimizations [16,17].

This study can be categorized into the following two aspects:

First, dimensionless transient governing equations for two widely adapted water-supply systems are developed: a reservoir pipeline surge tank valve reservoir system and a reservoir pump check valve pipeline surge tank valve reservoir system. Both a standard dimensionless expression for the surge tank and a simplified dimensionless lumped inertia are introduced, and solutions for combinations of various structures are developed.

Second, both frequency and time domain solutions for the two distinct systems are presented and compared. The effect of the surge tank on a specific section of the pipeline is investigated via the introduction of an integrated solution, and a comparison between different systems is performed to provide insights into the system response in terms of the contribution of isolation in the pressure response for a specific component.

Section 2 covered dimensionless governing equations and their solutions for two different water supply systems with surge tanks in the frequency domain. Section 3 provided frequency response functions and time domain responses for two systems in terms of point and integrated hydraulic impedance, respectively. Strengths and differences over other existing methods were discussed in Section 4. The summary of this study and future issues from this study were addressed in Section 5.

## 2. Materials and Methods

### 2.1. One-Dimensional Dimensionless Governing Equation with Steady Friction

The pressure head and flow rate variations in a pressurized pipeline system can be expressed using the partial differential equation (PDE) of continuity and momentum conservation as a function of two independent variables, i.e., time ($t$) and distance ($x$) [1]. Based on these PDFs, the dimensionless continuity and momentum equations can be developed as follows:

$$\frac{\partial V}{\partial t} + g\frac{\partial H}{\partial x} + \frac{fV|V|}{2DA^2} = 0 \tag{1}$$

$$\frac{a^2}{g}\frac{\partial V}{\partial x} + \frac{\partial H}{\partial t} = 0, \tag{2}$$

where $V$ is the mean velocity for the cross-sectional area $A$, $H$ the piezometric head, a the wave propagation speed, $g$ the gravitational acceleration, $f$ the Darcy–Weisbach friction factor, and $D$ the diameter.

This section may be divided into subheadings. It should provide a concise and precise description of the experimental results, their interpretation, as well as the experimental conclusions that can be drawn. By introducing the mean flow rate $Q$, which represents the mean velocity by multiplying the cross-sectional area, the dimensionless PDEs of momentum and mass conservation can be developed. The dimensionless independent variables for time and distance are defined as $\widehat{t} = at/L$ and $\widehat{x} = x/L$, respectively, where L denotes the length of the pipeline system. The dependent dimensionless variables are $\widehat{H} = gAH/(4aQ_o)$ for the pressure head and $\widehat{Q} = Q/Q_0$ for the flow rate, where $Q_0$ is the steady flow rate.

The dimensionless continuity and momentum equations can be derived as follows:

$$\frac{\partial \widehat{Q}}{\partial \widehat{t}} + \frac{\partial \widehat{H}}{\partial \widehat{x}} + \widehat{R} = 0 \tag{3}$$

$$\frac{\partial \widehat{H}}{\partial \widehat{t}} + \frac{\partial \widehat{Q}}{\partial \widehat{x}} = 0, \tag{4}$$

where $\widehat{R}$ is the dimensionless resistance, which can be estimated as $\widehat{R} = fLQ_o/(2DAa)$ under steady friction.

Applying the perturbation theory [1] to the dimensionless pressure head and flow rate expressed in Equations (3) and (4) yields the trigonometric relationship between the upstream and downstream dimensionless frequency ($\widehat{s}$) as follows:

$$\widehat{H_D} = \widehat{H_U}\cosh\widehat{\gamma}\widehat{x} - \widehat{Z_c}\widehat{Q_U}\sinh\widehat{\gamma}\widehat{x} \tag{5}$$

$$\widehat{Q_D} = -\frac{\widehat{H_U}}{\widehat{Z_c}}\sinh\widehat{\gamma}\widehat{x} + \widehat{Q_U}\cosh\widehat{\gamma}\widehat{x}, \tag{6}$$

where the dimensionless propagation constant, $\widehat{\gamma}$, can be expressed as

$$\widehat{\gamma} = \sqrt{\widehat{s}\left(\widehat{s} + \widehat{R}\right)}, \tag{7}$$

where $\widehat{s}$ is the dimensionless frequency.

*2.2. Dimensionless Hydraulic Impedance from Surge Tank to Joining Point*

The two-dimensional PDEs for the continuity and momentum in terms of time ($t$), axial distance ($x$), and radial distance ($r$) are expressed as follows:

$$\frac{\partial u}{\partial x} + \frac{g}{a^2}\frac{\partial h}{\partial t} = 0 \tag{8}$$

$$\frac{\partial u}{\partial t} + g\frac{\partial h}{\partial x} + \frac{v}{r}\frac{\partial}{\partial r}\left(r\frac{\partial u}{\partial r}\right) = 0 \tag{9}$$

where $u$ and $v$ are the axial velocity and dynamic viscosity, respectively.

By introducing dimensionless variables $\widehat{x} = x/L$; $\widehat{t} = at/L$; $\widehat{r} = r/R$; $\widehat{h} = gh/a^2$; and $\widehat{u} = u/a$, two-dimensional dimensionless equations for mass and momentum conservation can be derived as follows:

$$\frac{\partial \widehat{h}}{\partial \widehat{t}} + \frac{\partial \widehat{u}}{\partial \widehat{x}} = 0 \tag{10}$$

$$\frac{\partial^2 \widehat{u}}{\partial \widehat{r}^2} + \frac{1}{\widehat{r}} \frac{\partial \widehat{u}}{\partial \widehat{r}} - \frac{1}{S_a} \frac{\partial \widehat{u}}{\partial \widehat{t}} = \frac{1}{S_a} \frac{\partial \widehat{h}}{\partial \widehat{x}} \tag{11}$$

where $S_a = \nu L/aR^2$ is a dimensionless surge number. Based on the assumption of zero heat radiation, the integration of Equations (10) and (11) provides unsteady functions in the dimensionless frequency domain. The dimensionless pressure head and flow rate relationships between the upstream and downstream for Equations (10) and (11) can be expressed as follows:

$$\widehat{H_D}(\widehat{s}) = \widehat{H_U}(\widehat{s})\cosh\Gamma(\widehat{s}) - \widehat{Q_U}(\widehat{s})Z_c(\widehat{s})\sinh\Gamma(\widehat{s}) \tag{12}$$

$$\widehat{Q_D}(\widehat{s}) = -\frac{\widehat{H_U}(\widehat{s})}{Z_c(\widehat{s})}\sinh\Gamma(\widehat{s}) + \widehat{Q_U}(\widehat{s})\cosh\Gamma(\widehat{s}) \tag{13}$$

where $\widehat{H_D}(\widehat{s})$ and $\widehat{H_U}(\widehat{s})$ denote dimensionless complex heads; and $\widehat{Q_D}(\widehat{s})$ and $\widehat{Q_U}(\widehat{s})$ are dimensionless complex discharges for the downstream and upstream points, respectively.

The dimensionless characteristic impedance can be expressed as follows,

$$Z_c(\widehat{s}) = \sqrt{\frac{J_0(\sqrt{(\widehat{S}/S_a)i})}{J_0(\sqrt{(\widehat{S}/S_a)i}) - 2/(\sqrt{(\widehat{S}/S_a)i})J_1(\sqrt{(\widehat{S}/S_a)i})}} \tag{14}$$

where $J_0$ and $J_1$ are Bessel functions of 0th and 1st order, respectively. The dimensionless propagation constant can be expressed as follows:

$$\Gamma(\widehat{s}) = \widehat{x}\widehat{s}\sqrt{\frac{J_0(\sqrt{(\widehat{S}/S_a)i})}{J_0(\sqrt{(\widehat{S}/S_a)i}) - 2/(\sqrt{(\widehat{S}/S_a)i})J_1(\sqrt{(\widehat{S}/S_a)i})}} \tag{15}$$

### 2.3. Dimensionless Hydraulic Impedance from Surge Tank to Joining Point

The fluctuation in the flow rate from the surge tank to the main pipeline can be expressed as $q' = -A_S dh'/dt$, where $A_S$ is the surge tank area. The disturbance of the pressure head ($h'$) can be approximated as $h' = H_S e^{st}$, where $H_S$ is the steady pressure head in the surge tank. By defining the dimensionless fluctuations of pressure head and flow rate as $\widehat{h'} = gAh'/(aQ_o)$ and $\widehat{q'} = q'/Q_o$, respectively.

The dimensionless pressure head variation at the surge tank entrance can be expressed as

$$\widehat{h'} = \widehat{H_S}e^{i\widehat{s}\widehat{t}}, \tag{16}$$

where $\widehat{H_S} = \frac{gAH_S}{aQ_o}$.

The dimensionless flow rate fluctuation at the surge tank entrance can be expressed as

$$\widehat{q'} = \widehat{Q_S}e^{i\widehat{s}\widehat{t}}, \tag{17}$$

where $\widehat{Q_S} = -\frac{A_S aH_S}{LQ_o}i\widehat{s}$.

The hydraulic impedance at the surge tank outlet can be expressed as

$$\widehat{Z_S} = \frac{gAL}{A_S a^2 \widehat{s}} \tag{18}$$

If the surge tank is connected to the main pipeline by a connector, then the hydraulic impedance at the joining point of the main pipeline can be expressed as

$$\widehat{Z_J} = \widehat{H_j}(\hat{s}) / \widehat{Q_J}(\hat{s}) \tag{19}$$

The dimensionless pressure head and flow rate at the junction can be expressed, respectively, as follows:

$$\widehat{H_j}(\hat{s}) = \widehat{H_S}(\hat{s})\cosh\Gamma_c(\hat{s}) - \widehat{Q_S}(\hat{s})Z_{cc}(\hat{s})\sinh\Gamma_c(\hat{s}) \tag{20}$$

$$\widehat{Q_j}(\hat{s}) = -\frac{\widehat{H_S}(\hat{s})}{Z_{cc}(\hat{s})}\sinh\Gamma_c(\hat{s}) + \widehat{Q_S}(\hat{s})\cosh\Gamma_c(\hat{s}), \tag{21}$$

where $Z_{cc}(\hat{s}) = \sqrt{\dfrac{J_0(\sqrt{(\hat{S}/S_{ac})i})}{J_0(\sqrt{(\hat{S}/S_{ac})i})-2/(\sqrt{(\hat{S}/S_{ac})i})J_1(\sqrt{(\hat{S}/S_{ac})i})}}$, $S_{ac} = \nu L_c/aR_c{}^2$; $L_c$ and $R_c$ are the length and radius of the connector, respectively; and $\Gamma_c(\hat{s})$ $= \widehat{xs}\sqrt{\dfrac{J_0(\sqrt{(\hat{S}/S_{ac})i})}{J_0(\sqrt{(\hat{S}/S_{ac})i})-2/(\sqrt{(\hat{S}/S_{ac})i})J_1(\sqrt{(\hat{S}/S_{ac})i})}}$.

### 2.4. Dimensionless Lumped Inertia

Expression for accumulator in frequency domain had been approximated using the lumped inertia [1], which can be applied to surge tanks with short connectors. In this study, the existing lumped inertia expressions were extended into the dimensionless lumped inertia. The dimensionless lumped inertia model can widely address the effect of the surge tank. By introducing dimensionless variables for the lumped inertia, i.e., $\widehat{H} = \frac{gA}{aQ_o}H$, $\widehat{Q} = Q/Q_o$, $\widehat{l_c} = l_c/L$, the dimensionless relationship can be expressed as follows:

$$\widehat{H_D} = \widehat{H_U} - \widehat{l_c}i\hat{s}\widehat{Q_U} \tag{22}$$

Assuming that the dimensionless upstream flow rate is identical to that of the downstream flow for a short connector, the following dimensionless relationship can be derived:

$$\widehat{Z_D} = \widehat{Z_U} - \widehat{l_c}i\hat{s} \tag{23}$$

where $\widehat{Z_D}$ and $\widehat{Z_U}$ are the dimensionless hydraulic impedances downstream and upstream of the surge tank connector, respectively.

### 2.5. Development of Dimensionless Hydraulic Impedance for Two Different Systems

A surge tank is typically installed in a simple pipeline system, as shown in Figure 1. A hydraulic transient can be introduced through the abrupt closure of the downstream valve or the abrupt termination of the upstream pump. This can cause severe damage owing to either the overpressure or underpressure, which occasionally introduces cavitation.

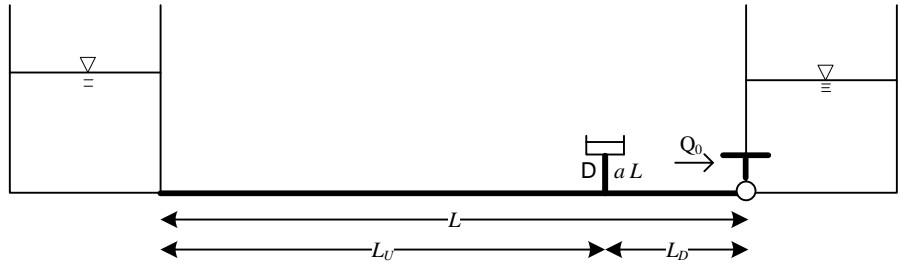

**Figure 1.** Schematic illustration of reservoir pipeline surge tank pipeline valve (R-P-ST-P-V) system.

The other widely used systems comprise a pump and check valve in the upstream section (see Figure 2), which allow the upstream pressure head to be increased when the pres-

sure head of the upstream reservoir is low, or when the distribution of the pipeline elevation requires an elevating additional pressure head to secure flow conveyance downstream.

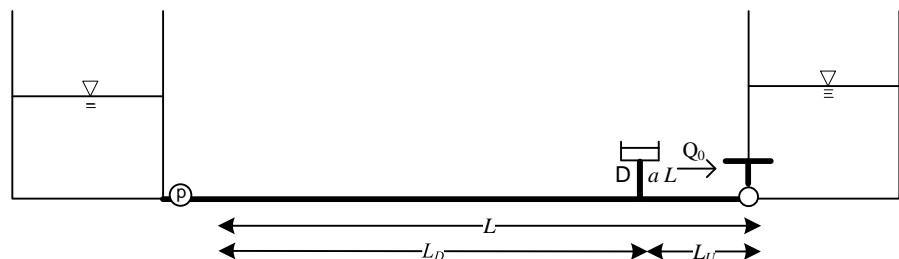

**Figure 2.** Schematic illustration of reservoir pump check valve pipeline surge tank pipeline valve (R-PP-CV-P-ST-P-V) system.

To generalize the system response feature in dimensionless space, the upstream length between the upstream reservoir and surge tank, $L_U$, can be converted into an upstream dimensionless length as $\widehat{L_U} = L_U/L$, and the dimensionless downstream length can be defined as $\widehat{L_D} = L_D/L$. If the driver of the hydraulic transient is the downstream valve in the pipeline system shown in Figure 1, then a dimensionless impedance from the upstream to downstream direction should be derived. The dimensionless hydraulic impedance upstream of the joining point is expressed as follows:

$$\widehat{Z}_{UJ} = -\widehat{Z}_c \tanh \widehat{\gamma} \widehat{L_U}, \tag{24}$$

where $\widehat{Z}_c = \widehat{\gamma}/\widehat{s}$ and $\widehat{\gamma} = \sqrt{\widehat{s}(\widehat{s} + \widehat{R})}$.

If the transient pressure head from downstream introduces a reversed flow in the upstream direction, then the check valve at the outlet of the pump can be instantly closed to protect the pump, and the system shown in Figure 2 can be changed in terms of the primary hydraulic impedance from the upstream boundary which can be expressed as follows:

$$\widehat{Z}_{UJ} = -\widehat{Z}_c \coth \widehat{\gamma} \widehat{L_U} \tag{25}$$

The dimensionless hydraulic impedance of the main pipeline downstream of the connecting element for the system shown in Figure 1 can be expressed as follows:

$$\widehat{Z_{DJ}} = \frac{-\widehat{Z}_c \tanh \widehat{\gamma} \widehat{L_U}}{1 - \widehat{Z}_c \tanh \widehat{\gamma} \widehat{L_U} / \widehat{Z_J}} \tag{26}$$

The corresponding dimensionless hydraulic impedance downstream of the connecting element in Figure 2 can be expressed as follows:

$$\widehat{Z_{DJ}} = \frac{-\widehat{Z}_c \coth \widehat{\gamma} \widehat{L_U}}{1 - \widehat{Z}_c \coth \widehat{\gamma} \widehat{L_U} / \widehat{Z_J}} \tag{27}$$

The dimensionless hydraulic impedance at the downstream valve can be expressed as follows:

$$\widehat{Z_{DV}} = \frac{\widehat{Z_{DJ}} - \widehat{Z}_c \tanh \widehat{\gamma} \widehat{L_D}}{1 - \widehat{Z_{DJ}} / \widehat{Z}_c \tanh \widehat{\gamma} \widehat{L_D}} \tag{28}$$

The dimensionless pressure head response from the downstream flow rate variation, such as the valve maneuver between the downstream valve and the connecting point for the surge tank, can be derived as follows:

$$\frac{\widehat{H_{xD}}}{\widehat{Q_{DV}}} = \widehat{Z_{DV}} \cosh \widehat{\gamma} \widehat{x_D} + \widehat{Z}_c \sinh \widehat{\gamma} \widehat{x_D}, \tag{29}$$

where $\widehat{x_D}$ is the dimensionless distance from the downstream valve to an upstream point up to the connecting point.

The dimensionless pressure head response between the connecting point and upstream reservoir can be derived as follows:

$$\frac{\widehat{H_{xU}}}{\widehat{Q_{DV}}} = \widehat{Z_{Uc}} \cosh \widehat{\gamma}\widehat{x_U} + \widehat{Z_c}\sinh\widehat{\gamma}\widehat{x_U}, \tag{30}$$

where $\widehat{x_U}$ is the dimensionless distance from the connecting point to a point up to the upstream reservoir. The dimensionless hydraulic impedance upstream of the connection point can be estimated as

$$\widehat{Z_{Uc}} = \frac{\widehat{Z_J} - \widehat{Z_{Dc}}}{\widehat{Z_{Dc}}\widehat{Z_J}}, \tag{31}$$

The dimensionless hydraulic impedance downstream of the connecting point can be estimated as

$$\widehat{Z_{Dc}} = \frac{\widehat{Z_{DV}} + \widehat{Z_c}\tanh\widehat{\gamma}\widehat{L_D}}{1 + \widehat{Z_{DV}}/\widehat{Z_c}\tanh\widehat{\gamma}\widehat{L_D}} \tag{32}$$

If the hydraulic transient is introduced from the pump stoppage of upstream reservoir, then the derivation should be performed from the reservoir downstream to the pump upstream, and the dimensionless hydraulic impedance at the pump can be expressed as

$$\widehat{Z_p} = \frac{\widehat{Z_{PU}} + \widehat{Z_c}\tanh\widehat{\gamma}\widehat{L_U}}{1 + \widehat{Z_{PU}}/\widehat{Z_c}\tanh\widehat{\gamma}\widehat{L_U}}, \tag{33}$$

where $\widehat{Z_{PU}} = (\widehat{Z_J}\widehat{Z_c}\tanh\widehat{\gamma}\widehat{L_D})/(\widehat{Z_J} - \widehat{Z_c}\tanh\widehat{\gamma}\widehat{L_D})$. The dimensionless pressure head response from the abrupt pressure change in the pump between the pump and connecting point can be derived as follows:

$$\frac{\widehat{H_{xu}}}{\widehat{H_p}} = \cosh \widehat{\gamma}\widehat{x_u} - \frac{\widehat{Z_c}}{\widehat{Z_p}}\sinh\widehat{\gamma}\widehat{x_u}, \tag{34}$$

where $\widehat{x_u}$ is the dimensionless distance from the upstream to downstream connecting point. The dimensionless pressure head response from the sudden pressure change in the pump between the connecting point and downstream valve can be evaluated as follows:

$$\frac{\widehat{H_{xd}}}{\widehat{H_p}} = \cosh \widehat{\gamma}\widehat{x_d} - \frac{\widehat{Z_c}}{\widehat{Z_{cd}}}\sinh\widehat{\gamma}\widehat{x_d}, \tag{35}$$

where $\widehat{x_d}$ is the dimensionless distance from the connecting point downstream, $\widehat{Z_{cd}} = \widehat{Z_{cu}}\widehat{Z_J}/(\widehat{Z_{cu}} + \widehat{Z_J})$, and the dimensionless hydraulic impedance upstream of the connection point can be expressed as

$$\widehat{Z_{cu}} = \frac{\cosh \widehat{\gamma}\widehat{L_U} - \widehat{Z_c}/\widehat{Z_p}\sinh\widehat{\gamma}\widehat{L_U}}{-\sinh\gamma\widehat{L_U}/\widehat{Z_c} + \cosh \widehat{\gamma}\widehat{L_U}/\widehat{Z_p}} \tag{36}$$

The dimensionless impedance approach provides an integrated pressure response along a specific pipeline section. The total pressure response between the connecting point and the downstream valve owing to valve closure can be expressed as follows:

$$\int_0^{\widehat{L_D}} \widehat{H_{xD}}/\widehat{Q_{DV}}d\widehat{x} = \frac{\widehat{Z_D}}{\widehat{\gamma}}\sinh\widehat{\gamma}\widehat{L_D} + \frac{\widehat{Z_c}}{\widehat{\gamma}}(\cosh \widehat{\gamma}\widehat{L_D} - 1) \tag{37}$$

The total pressure response between the upstream pump and connecting point due to valve closure can be expressed as follows:

$$\int_0^{\widehat{L_U}} \widehat{H_{xU}}/\widehat{Q_{DV}}\,d\hat{x} = \frac{\widehat{Z_U}}{\widehat{\gamma}}\sinh\widehat{\gamma}\widehat{L_U} + \frac{\widehat{Z_c}}{\widehat{\gamma}}(\cosh\widehat{\gamma}\widehat{L_U} - 1) \tag{38}$$

The total pressure response between the upstream pump and connecting point due to pump termination can be expressed as follows:

$$\int_0^{\widehat{L_U}} \widehat{H_{xu}}/\widehat{H_p}\,d\hat{x} = \frac{1}{\widehat{\gamma}}\sinh\widehat{\gamma}\widehat{L_U} - \frac{\widehat{Z_c}}{\widehat{Z_p}\widehat{\gamma}}(\cosh\widehat{\gamma}\widehat{L_U} - 1) \tag{39}$$

The total pressure response between the connecting point and downstream valve owing to pump termination can be expressed as follows:

$$\int_0^{\widehat{L_D}} \widehat{H_{xd}}/\widehat{H_p}\,d\hat{x} = \frac{1}{\widehat{\gamma}}\sinh\widehat{\gamma}\widehat{L_D} - \frac{\widehat{Z_c}}{\widehat{Z_{cd}}\widehat{\gamma}}(\cosh\widehat{\gamma}\widehat{L_D} - 1) \tag{40}$$

## 3. Results

In this study, the pipeline system used in a previous study was used to apply the proposed approach, as the widely used time domain modeling method (e.g., the method of characteristics) has been compared with the developed methods in multiple cases [16]. The total length of the pipeline was 150 m, and the diameters of the main and connected pipelines were 0.02 m. The pipeline lengths upstream and downstream from the surge tank were 145 and 5 m, respectively. The wave propagation speed in the pipeline was estimated at 1210.5 m/s The steady flow rate from the upstream reservoir or pump was $0.928 \times 10^{-4}$ m$^3$/s. The diameter of the surge tank was assumed to be 2 m, and the length of the connector was 0.5 m. The abrupt valve closure and termination of the check valve might introduce hydraulic transients. The maximum frequency for frequency domain modeling was terminated in 3812 rad/s and the number of fast Fourier transform for the conversion of the frequency domain to time domain response function is 32,768.

### 3.1. Frequency Response Function

The amplitudes of the dimensionless hydraulic impedances at the downstream valves of the systems shown in Figures 1 and 2 are presented in Figure 3. The solid line in Figure 3, denoted as R-P-ST-P-V, is the dimensionless frequency response function of the hydraulic impedance at the downstream valve in Figure 1. The dotted line in Figure 3 represents the corresponding frequency response of the system (R-PP-CV-P-ST-P-V), as shown in Figure 2. The overall shapes of the frequency response functions of the two systems were similar, showing positive peak responses at 43 and 132 and negative peaks at 87 and 178 in the dimensionless frequency domain. The difference in the primary resonance response from the lower frequency to the higher frequency increased from 43 to 46 for the four peak responses. The resonance pattern of the dimensionless hydraulic impedance can vary depending on the location of the surge tank. The difference in the frequency response between R-P-ST-P-V and R-PP-CV-P-ST-P-V was the phase shift, which can be explained by the difference in the transient generation positions between the two systems. Whereas the hydraulic transient for R-P-ST-P-V was generated from the downstream valve, that for R-PP-CV-P-ST-P-V was initiated from the upstream check valve.

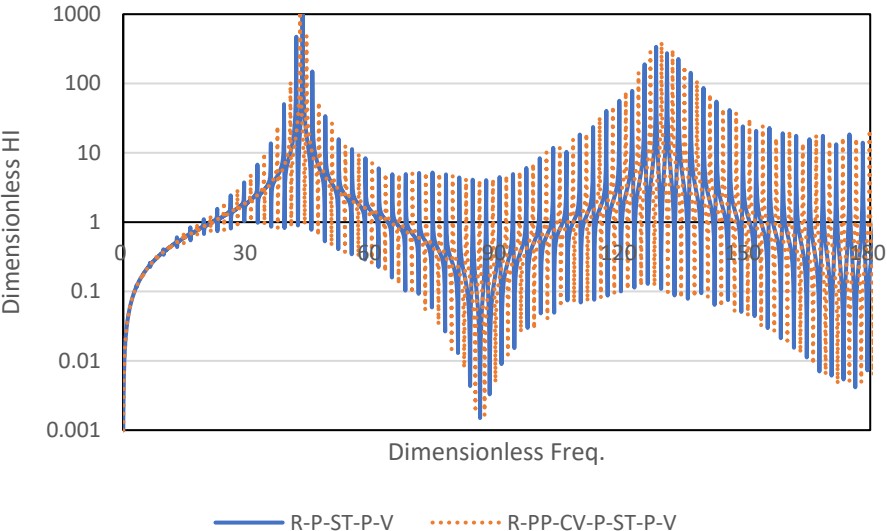

**Figure 3.** Amplitudes of dimensionless hydraulic impedance at downstream valve for reservoir pipeline surge tank pipeline-valve system (R-P-ST-P-V) in Figure 1 and that of reservoir pump check valve pipeline surge tank pipeline valve system (R-PP-CV-P-ST-P-V) shown in Figure 2.

Next, the frequency responses of the dimensionless expressions for the surge tank were analyzed. Figure 4 shows the amplitudes of the dimensionless hydraulic impedances at the downstream valve for the dimensionless surge tank expression (DLST) and dimensionless lumped inertia (DL LI) for the R-PP-CV-P-ST-P-V system. Even though slight differences in the amplitudes between the two distinct approaches were observed at several points, the distributions of the frequency responses were consistent with the development of the resonance pattern. This indicates that the dimensionless lumped inertia can be a reasonable approximation when the connector is short, which is applicable to most systems.

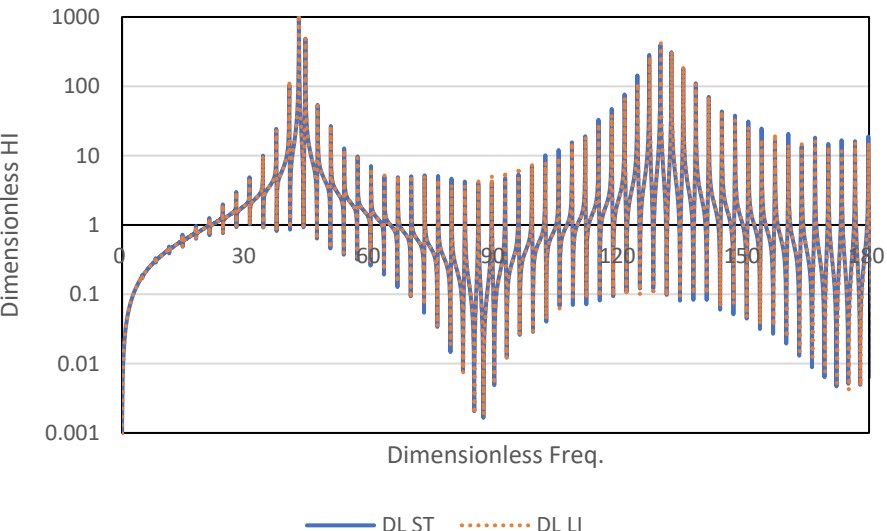

**Figure 4.** Amplitudes of dimensionless hydraulic impedance at downstream valve for reservoir pump check valve pipeline surge tank pipeline valve system (R-PP-CV-P-ST-P-V) using dimensionless surge tank (DLST) and dimensionless lumped inertia (DLLI).

The dimensionless frequency domain approach allows flexible analytical developments, such as obtaining integrated responses along any designated section of the pipeline system. This approach is particularly useful when the pipeline manager is required to estimate the potential pressure effect for a specific section and several designated points. Figure 5 shows the integrated dimensionless hydraulic impedances in the upstream section

of the surge tank, which were protected by the surge tank from the generated hydraulic transient originating from the downstream valve. The distributions of the integrated frequency responses of the two systems were similar. The mitigated frequency responses compared with those in Figure 3 indicate that both systems were well protected by the surge tank. The phase discrepancy in the integrated frequency response between the two systems can be explained by the difference in the pressure wave reflections from the distinct upstream boundaries.

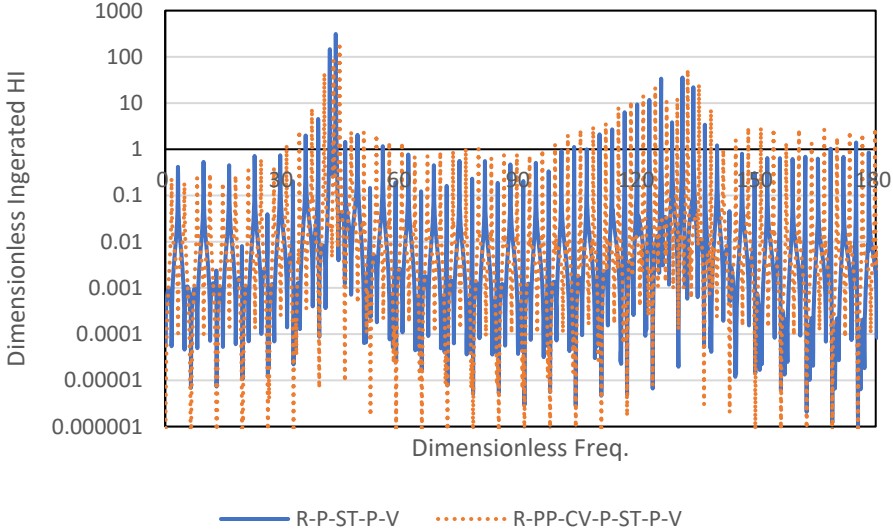

**Figure 5.** Amplitudes of integrated dimensionless hydraulic impedance along upstream section of surge tank for reservoir pipeline surge tank pipeline valve system (R-P-ST-P-V) in Figure 1 and that of reservoir pump check valve pipeline surge tank pipeline valve system (R-PP-CV-P-ST-P-V) shown in Figure 2.

### 3.2. Time Domain Pressure Response

The dimensionless frequency responses of the hydraulic impedance can be used to separate the corresponding time domain responses. Assuming that an abrupt flow rate change is introduced from the downstream valve via an instant valve closure, the flow rate variation can be convolved into the time domain response functions shown in Figures 3–5. Figure 6 shows the pressure responses at the downstream valve for R-P-ST-P-V and R-PP-CV-ST-P-V at the instant of the downstream valve closure. Figure 6 shows two distinct patterns in the pressure responses: a high-frequency component pattern between the downstream valve and surge tank, and a lower-frequency response pattern for all systems, which corresponds to the theoretical period of the pipeline system ($4L/a$). Both pressure responses showed perfectly matching results up to a dimensionless time of 2, which corresponded to a round travel time between different upstream boundaries and the downstream valve. The difference in the upstream boundary conditions between the two systems may have contributed to the phase difference in the pressure response from the dimensionless time 2, as shown in Figure 6.

As illustrated in Figure 4, the integrated pressure responses allow one to comprehensively evaluate the pipeline section. Figure 7 shows the normalized integrated pressure response for the upstream section of the system, as similarly shown in Figures 1 and 2.

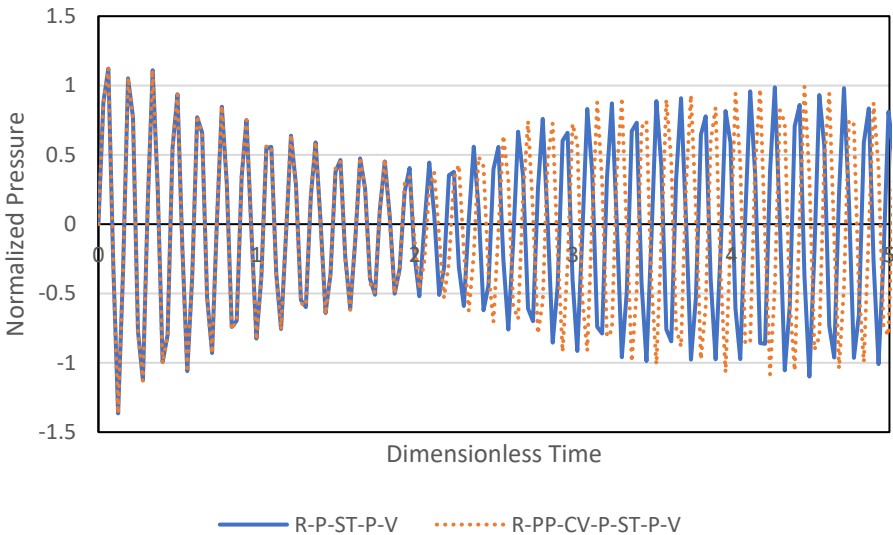

**Figure 6.** Normalized pressure responses at downstream valve due to instant valve closure of reservoir pipeline surge tank pipeline valve system (R-P-ST-P-V) in Figure 1 and that of reservoir pump check valve pipeline surge tank pipeline valve system (R-PP-CV-P-ST-P-V) in Figure 2.

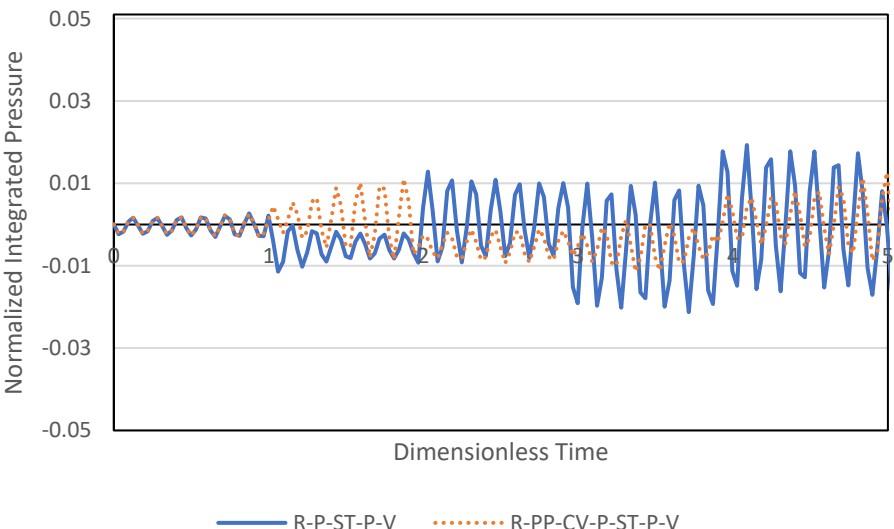

**Figure 7.** Normalized integrated pressure responses for upstream section of surge tank due to instant valve closure for reservoir pipeline surge tank pipeline valve system (R-P-ST-P-V) in Figure 1 and that of reservoir pump check valve pipeline surge tank pipeline valve system (R-PP-CV-P-ST-P-V) in Figure 2.

The scales of the integrated pressure response were substantially smaller than those of the pressure at the downstream valve, thereby indicating a significant surge-arresting effect of the surge tank. The integrated pressure responses for R-P-ST-P-V and R-PP-CV-P-ST-P-V matched perfectly with each other up to a dimensionless time of 1, which corresponded to the wave travel time from the downstream valve to the upstream boundary. As shown in Figure 7, the difference in the integrated responses was initiated from dimensionless time 1, and the pattern of R-P-ST-P-V reflected the mitigation from the upstream reservoir, whereas that of R-PP-CV-P-ST-P-V was doubled increasingly between dimensionless times 1 and 2 owing to the dead-end boundary from the check valve. The development of integrated pressure response patterns was evident at each dimensionless time interval, thus indicating that the dimensionless time defined in this study is a useful measure for characterizing pressure response patterns based on the boundary conditions and operational practices for pipeline systems.

## 4. Discussion

The solutions of a partial differential equation (PDE) for water hammer events have been explored through the conversion of PDE into an ordinary differential equation, namely MOC, in most applications. However, discretization approaches (e.g., MOC) have a critical restriction in representing the system. The requirement of Courant number related to wave speed and time–space interval in modeling introduces substantial computation costs and accurate representation of system dimensions. The location of the surge tank, check valve, pump and the dimensions of the surge tank should be fitted into the Courant number condition, which needs either assumptions or substantial computational costs. However, the introduction of a general solution in the frequency domain not only solved many existing issues in transient analysis but also provided the resonance response feature. The solutions of a hydraulic transient in the dimensionless space extend the applicability of the proposed method in general purpose if the system layout is identical. Furthermore, the evaluation of hydraulic impedance in dimensionless frequency provides a comprehensive understanding of system behavior as well as a universal solution in frequency response. As the dimensionless variables of system features, such as the location of the surge tank and dimensions, are changed, the frequency responses (Figures 3 and 5) provide an intuitive understanding of resonance response between pipeline and hydraulic structures. One additional strength of the frequency domain approach is the integrated expressions (Equations (37)–(40)). The direct integration in the frequency domain provides the evaluations of pressure or flow rate along a specific section. This means that the objective function for the optimization of hydraulic structure can be formulated not only from a point evaluation perspective but also through spatial integration, which can be useful for surge protection for vulnerable sections of the pipeline system.

## 5. Conclusions

Frequency domain responses due to the surge tank in pipeline systems with and without pumping stations and check valves were investigated. The frequency response of the hydraulic impedance at the downstream valve indicated that the resonance of the system can be explained by the pipeline length and the locations of the pipeline structures, such as the surge tanks, pumps, and check valves. The dimensionless approach provides response features in dimensionless time and amplitude, thus providing an intuitive explanation for the travel time, phase, and magnitude of the pressure response with the referenced scale. The difference in the frequency domain response was consistent with that in the time domain response, which depends on the system layout. The integrated response of the pressure along the pipeline segment allows one to comprehensively evaluate the pressure at a designated section. The difference in the time domain response between the two systems indicates that the amplification and mitigation originated from the boundary conditions with interaction with the surge tank. The dimensionless time and hydraulic impedance provide an intuitive understanding of the system response in the context of system dimensions, both in time and space. The characterization of the system behavior can be obtained through the evaluation of isolated hydraulic components in a dimensionless space. Future studies can be conducted to address the dynamic features of pumps and nonlinear valve behaviors. Additionally, the optimization of specific external disturbances (such as surges) can be considered in the design of surge protection devices.

**Funding:** This research was funded by the National Research Foundation of the Republic of Korea, grant number 2022R1A4A5028840.

**Data Availability Statement:** The data that support the findings of this study are available on request from the corresponding author, S. Kim.

**Conflicts of Interest:** The author declares no conflict of interest.

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
