# Peer review of "Dimensionless Pressure Response Analysis for Water Supply Pipeline Systems with or without Pumping Station"

_water, doi:10.3390/w15162934_

Round 1

Reviewer 1 Report

Please read the attachment. Thank you.

Author Response

⎯ Line 7: please remove “Affiliation 2; [email protected]”.

Thanks. this part I was removed.

⎯ Please add a manuscript outline at the end of the introduction section.

In order to address reviewer’s comment following contexts were added.

“The materials and method section covered dimensionless governing equations and their solutions for two different water supply systems with surge tank in frequency domain. Result section provided frequency response functions and time domain responses for two systems in terms of point and integrated hydraulic impedance, respectively. Strengths and differences over other existing methods were discussed in discussion section. Summary and future issues for this study were addressed in conclusion section.”

⎯ All equations should be mentioned or explained in the text.

Several words in context were changed to addressed reviewer’s comment. The manuscript included all necessary definitions for all equations.

⎯ Figure 2: there is an unnecessary line. Please remove.

Thanks. the unnecessary line in Figure 2 was removed.

⎯ Literature review: Please cite the references for the pump optimization. The following work could be helpful.

+ CFD Analysis and Optimum Design for a Centrifugal Pump Using an Effectively Artificial Intelligent Algorithm

+Centrifugal Pump Design: An Optimization

Thanks. Following context was added with citations of two paper.

Actually, computational fluid dynamic analyses had been performed to improve the design of centrifugal pump using multiple objective optimization [17-18].

  1. Wang, C.-N.; Yang, F.-C.; Nguyen, V.T.T.; Vo, N.T.M. CFD Analysis and Optimum Design for a Centrifugal Pump Using an Effectively Artificial Intelligent Algorithm. Micromachines 2022, 13, 1208. https://doi.org/10.3390/mi13081208
  2. Nguyen, V.T.T.; Vo, N.T.M. Centrifugal Pump Design: An Optimization. The Eurasia Proceedings of Science, Technology, Engineering & Mathematics 2022, 17, pp. 136-151.

⎯ Please provide a flowchart of the study process.

The main contribution of this paper is not algorithm but mathematical development of frequency domain solution for two widely used water supply systems in dimensionless space. Therefore, the flowchart of this work is too simple to present. Coding of solutions of governing equations and running program to obtain results. There is no calibration and feeds back from measurements because main content of this paper is mathematical development of solutions. Thanks for your understanding.

⎯ The literature review is limited (only 13). Please add more. The authors could find the proposed works in Water Journal or other similar journals for the related word to enrich your literature review.

In order to address reviewers comment following literature were added as follows,

“Many studies showed that the control of waterhammer or design of surge tank in pipeline systems can be performed based on MOC basis [11-13].

  1. Bhattarai, K.P.; Zhou, J.; Palikhe, S.; Pandey, K.P.; Suwal, N. Numerical Modeling and Hydraulic Optimization of a Surge Tank Using Particle Swarm Optimization. Water 2019, 11, 715. https://doi.org/10.3390/w11040715
  2. Mahmoudi-Rad, M.; Najafzadeh, M. Effects of Surge Tank Geometry on the Water Hammer Phenomenon: Numerical Investigation. Sustainability 2023, 15, 2312. https://doi.org/10.3390/su15032312
  3. Wan, W.; Wang, Y.; Chen, X.; Zhan, H.; Wang, T.; Zhang, B. Investigation of partially expanded surge tank with self‑adaptive auxiliary system controlling waterhammer in pipelines. Engineering Science and Technology, An International Journal, 2023, 40, 101379‑. https://dx.doi.org/10.1016/j.jestch.2023.101379”

Constructive questions:

  1. How does the location of the surge tank and check valve in the pipeline system with a pump impact the system resonance and pressure response in the frequency domain? Are there specific distances or configurations that lead to improved surge protection and system stability?

In this study, the location of hydraulic device impact was addressed through analytical derivations such as Equations (24)-(40) depending on dimensionless various variables. For your second question, it depends on specific system’s dimension and feature. This work generalized these impacts in the dimensionless space in terms of analytical expression.

  1. Can the dimensionless frequency and time domain analysis be extended to consider variations in water demand or flow rates within the water supply systems? How might these variations affect the observed pressure response patterns and the efficiency of surge protection mechanisms?

This is related to the basic assumption of steady oscillatory flow analysis of frequency domain analysis (Wyile and Streeter, 1993). If flow and pressure variation is associated with transient based on constant reservoir (energy source), then its can be simulated accurately. However, relaxation of this assumption looks out of scope of this paper, because this paper handle surge arresting issue based on transient generation of valve and pump action.

Wylie, E.B.; Streeter V.L. Fluid transients in systems 3rd ed. Prentice-Hall International: London, UK, 1993; pp. 37–79.

  1. In the context of practical applications, how can the insights gained from this study be utilized to optimize the design and operation of water supply systems? Are there any guidelines or best practices that engineers can implement to enhance system performance based on the phase difference in frequency response distribution between the simple pipeline system and the system with a pump and check valve?

Thanks. This is an important question. This work is one of mathematical developments to find general solution(dimensionless) for this question. If specific dimensions for real life systems are used and converted from dimensionless variable to real number then the result can be used for practical application. Optimization with possible scenarios of transient provide useful guideline for design of system. Conclusion address this issue as follows,

“The dimensionless time and hydraulic impedance provide an intuitive understanding of the system response in the context of system dimensions, both in time and space. The characterization of the system behavior can be obtained through the evaluation of isolated hydraulic components in a dimensionless space. Future studies can be conducted to address the dynamic features of pumps and nonlinear valve behaviors. Additionally, the optimization of specific external disturbances (such as surges) can be considered in the design of surge-protection devices.”

Thanks for your constructive comments. We will develop further and elaborate this work for more practical application in following work. This work is one of important foundations for this general objective.

Reviewer 2 Report

This is a very interesting contribution to the field of transient flow hydraulics, presenting the frequency domain approach to classical problems. Overall, the manuscript needs to improve the development of a knowledge gap it addresses in the introduction, and I left a few additional comments in the attached file. I believe that upon addressing these, the manuscript will be ready for publication.

Author Response

Line 43 Not a common terminology

  • This part is rephrased as “surge protection”

Line 49 Best justification for frequency domain method versus other methods

  • The only way to catch resonance feature of pipeline system is frequency domain analysis. However, impedance method provided additional advantage that provide time domain response as well. Therefore, this context was revised as follows;
  • “The application of the impedance method demonstrates the potential of the frequency-domain approach in the context of resonance characterization for the design of hydraulic structures via transient analysis as well as its complementary time domain response [15]”.

Line 64-67 Unclear if this is the knowledge gap, needs to be better stated if this is it.

  • This sentence is for motivation of this study. The pipeline system equipped with pumping station had not been explored using dimensionless approach yet. In order to address reviewer’s point following context was added

“Actually, computational fluid dynamic analyses had been performed to improve the design of centrifugal pump using multiple objective optimization [17-18]. However, proper consideration of pump with check valve and surge tank system had not been explored yet.”

Line 81 Adding figures to illustrate

  • This section is about the mathematical developments for dimensionless solution in frequency domain based on widely known mass and momentum equations of pressurized pipeline system. Figure is not about the partial differential equations and its solution development in frequency domain.

Line 106 perturbation theory add source

  • Reference Wyile and Streeter was added as [1]

Line 109 Are there function x-hat

  • Correct, as equation (5) showed H_hat is function of x_hat. x-hat is dimensionless length defined in line 110.

Line 145 Is Q0 dividing?

  • It is corrected as .

Line 203 smaller font

  • Font is corrected.

Line 237 could another variable?

  • The notation of this variable is changed as .

Line 266. MOC comparision?

  • Comparisons between MOC and impedance method had been done in many author’s previous papers. One of them is [16]. There many comparisons between two methods in many papers. It may not be worth while to revisit this issue.
  • Kim, S.H.; Choi, D. Dimensionless Impedance Method for General Design of Surge Tank in Simple Pipeline Systems. Energies, 2022, 15(10), pp 3603, https://doi.org/10.3390/en15103603.

Line 267. Length of line upstream and downstream

  • This part is addressed as follows
  • “The pipeline lengths of upstream and downstream from surge tank were 145 and 5 m, respectively.”

Line 319. Amplitudes

  • It exceeds limits 10^2 and 10^-5 slightly. Figure 5 was revised. Thanks.

Line 364 Discussion

  • Some context was removed to avoid redundancy.

Thanks for your constructive comments.

Reviewer 3 Report

Paper water-2533502 “Dimensionless Pressure Response Analysis for Water Supply Pipeline Systems with or without Pumping Station”

Comments

This study focuses on the dimensionless pressure response analysis for water supply pipeline systems with or without pumping station. I think the paper fits well the scope of the journal and addresses an important subject. However, a number of revisions are required before the paper can be considered for publication. There are some weak points that have to be strengthened. Below please find more specific comments:

*Abstract: I think the abstract reads well for the most part. The only suggestion I have is to include a sentence or two highlighting the outcomes of this research.

*Keywords: The keywords seem to be adequate. Some keywords seem to be kind of long though (e.g., Pressure Response of Waterhammer, Pumping Station with Check Valve). Please make the keywords more concise.

*The introduction section could benefit from more statistical information to better highlight the importance of the primary subject at hand.

*Literature review: The literature review seems kind of short for the most part. Please double check for the most recent and relevant studies published over the last 2-3 years. There should be a sufficient coverage of the most recent and relevant studies on this topic. The authors should carefully conduct the literature review, clearly define the existing gaps in the state of the art, and state how the present study contributes to the state of the art.

*Please make sure that all the adopted assumptions are supported by the relevant references. This will help justifying the adoption of these assumptions.

*The presentation of the proposed methodology seems to be adequate. Please make sure that the adopted mathematical relationships are supported by the relevant references where appropriate. There are many mathematical relationships listed in the manuscript. However, no supporting references are provided by the authors. This should be addressed in the revised version of the manuscript to provide the supporting references that could be of interest to some readers.

*Please provide more details regarding the input data used throughout this study. More supporting references would be helpful to justify the data selection.

*The manuscript contains quite a lot of figures. Please double check and try to provide a more detailed description of these figures where appropriate to make sure that the future readers will have a reasonable understanding of what these figures represent.

*The conclusions section should expand on limitations of this study and future research needs. I suggest listing the bullet points.

Author Response

Abstract: I think the abstract reads well for the most part. The only suggestion I have is to include a sentence or two highlighting the outcomes of this research.

  • In order to address reviewer’s comment following sentence was added.
  • “The driven responses in frequency and time domain can be the important basis for optimum design and operation conditions of water supply systems in dimensionless space.”

*Keywords: The keywords seem to be adequate. Some keywords seem to be kind of long though (e.g., Pressure Response of Waterhammer, Pumping Station with Check Valve). Please make the keywords more concise.

  • This is revised as follows;

Keywords: Waterhammer; Pipelines with Surge Tank; Dimensionless Responses; Pump with Check Valve

*The introduction section could benefit from more statistical information to better highlight the importance of the primary subject at hand.

  • The main contribution of this paper is analytical derivations for dimensionless frequency domain solution. Statistics seems not related to mathematical development. Thanks for your understanding.

*Literature review: The literature review seems kind of short for the most part. Please double check for the most recent and relevant studies published over the last 2-3 years. There should be a sufficient coverage of the most recent and relevant studies on this topic. The authors should carefully conduct the literature review, clearly define the existing gaps in the state of the art, and state how the present study contributes to the state of the art.

  • Thanks for suggestion, we review recently published related papers and added as follows;
  • Bhattarai, K.P.; Zhou, J.; Palikhe, S.; Pandey, K.P.; Suwal, N. Numerical Modeling and Hydraulic Optimization of a Surge Tank Using Particle Swarm Optimization. Water 2019, 11, 715. https://doi.org/10.3390/w11040715
  • Mahmoudi-Rad, M.; Najafzadeh, M. Effects of Surge Tank Geometry on the Water Hammer Phenomenon: Numerical Investigation. Sustainability 2023, 15, 2312. https://doi.org/10.3390/su15032312
  • Wan, W.; Wang, Y.; Chen, X.; Zhan, H.; Wang, T.; Zhang, B. Investigation of partially expanded surge tank with self‑adaptive auxiliary system controlling waterhammer in pipelines. Engineering Science and Technology, An International Journal, 2023, 40, 101379‑. https://dx.doi.org/10.1016/j.jestch.2023.101379
  • Wang, C.-N.; Yang, F.-C.; Nguyen, V.T.T.; Vo, N.T.M. CFD Analysis and Optimum Design for a Centrifugal Pump Using an Effectively Artificial Intelligent Algorithm. Micromachines 2022, 13, 1208. https://doi.org/10.3390/mi13081208
  • Nguyen, V.T.T.; Vo, N.T.M. Centrifugal Pump Design: An Optimization. The Eurasia Proceedings of Science, Technology, Engineering & Mathematics 2022, 17, pp. 136-151.

*Please make sure that all the adopted assumptions are supported by the relevant references. This will help justifying the adoption of these assumptions.

  • The steady oscillatory flow assumption was most important one from the Reference (Wyile and Streeter 1993). Thanks.
  • Wylie, E.B.; Streeter V.L. Fluid transients in systems 3rd ed. Prentice-Hall International: London, UK, 1993; pp. 37–79.

*The presentation of the proposed methodology seems to be adequate. Please make sure that the adopted mathematical relationships are supported by the relevant references where appropriate. There are many mathematical relationships listed in the manuscript. However, no supporting references are provided by the authors. This should be addressed in the revised version of the manuscript to provide the supporting references that could be of interest to some readers.

  • The dimensionless development for pipeline pressure system in frequency domain was performed by author’s previous works [15-16] which was based on the assumption of steady oscillatory flow assumption [1].
  • Kim, S.H.; Design of surge tank for water supply systems using the impulse response method with the GA algorithm. J. Mechanical Science and Technology, 2010, vol. 24(2), pp. 629-636.
  • Kim, S.H.; Choi, D. Dimensionless Impedance Method for General Design of Surge Tank in Simple Pipeline Systems. Energies, 2022, 15(10), pp 3603, https://doi.org/10.3390/en15103603.
  • Wylie, E.B.; Streeter V.L. Fluid transients in systems 3rd ed. Prentice-Hall International: London, UK, 1993; pp. 37–79.

*Please provide more details regarding the input data used throughout this study. More supporting references would be helpful to justify the data selection.

  • The input data for this paper is mostly identical to those published by author [15-16]. Those papers were cited widely. In order to address reviewer’s concern, following sentences were added further,

“The pipeline lengths of upstream and downstream from surge tank were 145 and 5 m, respectively. ~~. Maximum frequency for frequency domain modeling was terminated in 3812 rad/s and the number of fast Fourier transform for the conversion of frequency domain to time domain response function is 32768.”

*The manuscript contains quite a lot of figures. Please double check and try to provide a more detailed description of these figures where appropriate to make sure that the future readers will have a reasonable understanding of what these figures represent.

  • Figures presented in paper can be classified into three.
  • First is the schematic of systems such as Fig. 1 and Fig 2. Second is the frequency response function at point scale and integrated form for two different systems. Third is time domain responses for two different system. The purpose of these figures are illustrations of result showing resonance and time domain response in dimensionless space both time and pressure. The understanding of these response pattern can be achieved through the better understanding of response function in formulations presented in paper. Explaining these variables looks kind of redundancy because accurate mathematical definitions are given in the method section. Mathematical formulation itself seems the best explanation.

*The conclusions section should expand on limitations of this study and future research needs. I suggest listing the bullet points.

  • The developments in this paper should be elaborated further through implementation of more complicate nonlinear behavior (e.g. nonlinear valve feature and pumping station) for application of field systems. Actually, this part is already addressed as follows;

“Future studies can be conducted to address the dynamic features of pumps and nonlinear valve behaviors. Additionally, the optimization of specific external disturbances (such as surges) can be considered in the design of surge-protection devices.”

Thanks for your constructive comments.

Round 2

Reviewer 3 Report

The authors took seriously my previous comments and made the required revisions in the manuscript. The quality and presentation of the manuscript have been improved. Therefore, I recommend acceptance.